# Interaction between the *PNPLA3* Gene and Nutritional Factors on NAFLD Development: The Korean Genome and Epidemiology Study

**DOI:** 10.3390/nu15010152

**Published:** 2022-12-28

**Authors:** Sooyeon Oh, Jooho Lee, Sukyung Chun, Ja-Eun Choi, Mi Na Kim, Young Eun Chon, Yeonjung Ha, Seong-Gyu Hwang, Sang-Woon Choi, Kyung-Won Hong

**Affiliations:** 1Chaum Life Center, CHA University School of Medicine, Seoul 06062, Republic of Korea; 2Department of Gastroenterology, CHA Bundang Medical Center, CHA University School of Medicine, Seongnam 13496, Republic of Korea; 3Healthcare R&D Division, Theragen Bio Co., Ltd., Suwon 16229, Republic of Korea

**Keywords:** non-alcoholic fatty liver disease, sodium, kimchi, fermented vegetable

## Abstract

Genetic and nutritional factors contribute to the development of non-alcoholic fatty liver disease (NAFLD); however, gene–diet interactions in NAFLD development are poorly understood. In this case–control study, a large dataset from the Korean Genome and Epidemiology Study cohort (*n* = 72,299) comprising genomic data, medical records, social history, and dietary data was used. We investigated the interactions between the *PNPLA3* rs738409 genotype and nutritional factors and their possible effect on the risk of NAFLD development in 2950 patients with NAFLD and 12,907 controls. In the *PNPLA3* risk allele group, high protein, fat, sodium, phosphorus, niacin, and vitamin B6 intakes were associated with a decreased risk of NAFLD. In the non-risk allele group, only high fat intake was associated with a decreased risk of NAFLD. Among these nutrients, high sodium intake had a significant protective interaction with the *PNPLA3* genotype against NAFLD (*p* = 0.002). Among salty foods, only kimchi had a significant protective effect against the *PNPLA3* genotype (*p* = 0.012). Thus, the *PNPLA3* genotype is differentially associated with nutritional factors. In particular, it interacts with kimchi, a fermented vegetable dish. Therefore, fermented vegetables may serve as a tailored therapeutic food for people with the *PNPLA3* risk allele.

## 1. Introduction

Non-alcoholic fatty liver disease (NAFLD) is one of the most common liver diseases worldwide, and its prevalence continues to increase [1]. As a spectrum of liver disease that begins with simple steatosis, NAFLD is considered a hepatic manifestation of metabolic syndrome. Owing to an increase in obesity, the total number of NAFLD cases is projected to increase by 0–30% between 2016–2030 [2]. This is a public health concern because NAFLD can progress to non-alcoholic steatohepatitis, cirrhosis, and ultimately hepatocellular carcinoma (HCC). Statistics have shown that the incidence of HCC associated with NAFLD is increasing [3]. If this trend persists, NAFLD will become the principal cause of liver transplantation and liver-related mortality in the coming decades [4]. 

Traditionally, a sedentary lifestyle with a high calorie intake but low energy output has been considered the major cause of hepatic fat accumulation. Recent investigations have provided new insights into the pathogenesis of NAFLD. Genome-wide association studies (GWAS) have discovered that some single nucleotide polymorphisms (SNPs) increase the risk of NAFLD, including patatin-like phospholipase domain-containing 3 (*PNPLA3)*, transmembrane 6 superfamily member 2 (*TM6SF2*), sorting and assembly machinery component 50 (*SAMM-50*), farnesyl diphosphate farnesyl transferase I (*FDFT1*), collagen type XIII alpha 1 (*COL13A1*), neurocan (*NCAN*), glucokinase regulatory protein (*GCKR*), membrane-bound O-acyltransferase domain-containing protein 7 (*MBOAT7*), apolipoprotein C3 (*APOC3*), sterol regulatory element binding transcription factor 2 (*SREBF2*) rs133291, membrane-bound O-acyltransferase domain-containing 7 transmembrane channel-like 4 (*MBOAT7-TMC4*) rs641738, 17β-hydroxysteroid dehydrogenase type 13 (*HSD17B13*), and serpin family member 1 (*SERPINA1*) [5,6,7,8,9]. Among these, the association between *PNPLA3* and NAFLD risk is the most robustly observed [6]. The *PNPLA3* rs738409 C > G SNP, which leads to the replacement of isoleucine with methionine at position 148 (I148M), contributes to the development of NAFLD by promoting triglyceride synthesis and accumulation in hepatocytes [10]. The effect of this SNP was even observed in subjects with “lean NAFLD”, a condition where non-obese individuals are predisposed to develop NAFLD [11].In these circumstances, where genetic factors directly affect the pathogenesis but no specific drug therapy exists, the role of diet becomes crucial. Thus far, only a few studies have suggested the role of gene–diet interactions in NAFLD development. One such study discovered that a high intake of carbohydrates, isoflavones, and methionine increased the risk of hepatic fibrosis in NAFLD in a *PNPLA3* genotype-dependent manner [12]. Another study showed that the minor allele of the haplotype in the 22q13 loci increased the risk of NAFLD via interaction with a high carbohydrate intake, including high consumption of noodles and meat [13]. Therefore, certain dietary patterns should be avoided by individuals with risk genotypes, and certain dietary patterns could potentially attenuate the genetic influence. These findings justify the idea of precision nutrition, a diet tailored to one’s genetic make-up [10,12,13,14,15]. In addition, dietary factors modulate the crosstalk between the gut microbiome and the liver through portal circulation [16,17]. Therefore, diet can also affect the gut-liver axis that plays a critical role in the development of NAFLD.In Korea, the prevalence of NAFLD (32.87–42.9%) exceeded the global average (25.24%) [9,18]. It is expected to increase in the future owing to a Westernized diet, lack of exercise, and increase in obesity and type 2 diabetes [9]. The Korean diet is based on steamed rice served with small side dishes; these side dishes are prepared mainly with vegetables, and less frequently with meat, poultry, or fish. Fermented vegetables such as kimchi are also used quite frequently. Furthermore, the traditional Korean diet uses whole mixed grains and beans; thus, it is known to be healthy, having a low glycemic index, low cholesterol, and high fiber content [19]. However, these traditional benefits have been overshadowed by an excessive consumption of refined rice, along with an increased consumption of Western foods [20]. 

In this study, we hypothesized that a distinctive gene–diet interaction may exist between *PNPLA3* and NAFLD. We planned to test this interaction in the Korean population. The majority of Koreans belong to one ethnicity, called han minjok [21], which results in a genetic and dietary homogeneity that is beneficial for genetic and dietary studies. We used a large Korean cohort (Korean Genome and Epidemiology Study [KoGES] cohort) dataset obtained from the Korea Biobank Project (KBP) database. 

## 2. Materials and Methods

### 2.1. Study Population and the Definition of NAFLD 

In this case–control study, we received a dataset from the KoGES City cohort (*n* = 72,299) from KBP [22] that was collected from January 2004 to December 2012. The dataset included genomic data, medical records, social history, and dietary data. Patients with NAFLD were screened for the case group if they had a medical record of fatty liver disease but had neither viral hepatitis nor alcohol consumption exceeding 210 g or more per week for men and 140 g or more per week for women. Subjects without NAFLD or other underlying diseases were screened for the control group. Subjects missing medical records of fatty liver disease were excluded. 

### 2.2. Dietary Assessment

For each nutrient, a daily intake either above or below the recommended daily value was defined as high intake or low intake, respectively. The recommended daily values used in this study are presented in Appendix A. The median value adjusted by the total energy was taken as the cut-off, and a higher intake above the median was defined as a high intake value. 

For the dietary assessment, a semi-quantitative food-frequency questionnaire (FFQ) [23,24,25] comprising 103 items was developed for the KoGES data [26]. Participants reported the frequency and number of foods consumed over the past year using the FFQ. The intake of a food item three times or more per week was defined as high intake and less than three times a week as low intake. The KoGES data also provided information on the intake of 23 nutrients. 

### 2.3. Genome-Wide Genotyping 

The genotype data were provided by the Center for Genome Science, Korea National Institute of Health and were processed using the Korea Biobank Array (Affymetrix, Santa Clara, CA, USA). The experimental results of the Korea Biobank Array were filtered using the following quality control criteria: call rate > 97%, minor allele frequency > 1%, and Hardy–Weinberg equilibrium, *p* < 1 × 10^−5^. After quality control filtering, the experimental phenotypes were used to analyze the genotype datasets from the 1000 Genome Phase 1 and 2 Asian panels. GWAS identified 7,975,321 SNPs on chromosomes 1–22. The SNP genotype (*PNPLA3* rs738409) of participants was extracted from the Korea Biobank array (referred to as KoreanChip), which was optimized for the Korean population to demonstrate the findings of blood biochemical traits through GWAS [27].

### 2.4. Statistical Analysis

We performed GWAS in our KoGES City cohort (*n* = 72,299) to identify the genetic indicators of NAFLD. The genetic risk of the *PNPLA3* rs738409 SNP for NAFLD development was tested using additive, dominant, and recessive models. We compared the baseline characteristics between participants with and without NAFLD using the Student’s t-test for continuous variables and Pearson’s chi-squared test for categorical variables. Furthermore, the association between the *PNPLA3* rs738409 SNP and NAFLD was assessed using Pearson’s chi-squared test and logistic regression analysis. The association was adjusted for age, sex, body mass index, smoking status, and alcohol intake. The influence of dietary factors on the *PNPLA3* rs738409 SNP risk allele and non-risk allele groups was investigated, as was the interaction between dietary factors and the *PNPLA3* rs738409 SNP. All genetic association tests were conducted using PLINK version 1.9 (https://www.cog-ge-nom-ics.org/plink) [28], and the phenotype characteristics were analyzed using the SPSS (IBM SPSS Statistics Inc., New York, NY, USA) [29] and R statistical software (v4.1.2; R Core Team) [30]. Differences were considered statistically significant at two-sided *p* values < 0.05.

## 3. Results

In the KoGES City cohort (*n* = 72,299), the *PNPLA3* rs738409 *G* allele frequency was 0.41. Among the subjects (*n* = 72,299), those with missing data (*n* = 10,676) were excluded. In a review of medical records, subjects with fatty liver disease but without viral hepatitis were screened (*n* = 3568). Among them, participants with alcohol consumption exceeding the predefined criteria were excluded (*n* = 618). Among the controls (*n* = 58,055), subjects with underlying diseases were excluded (*n* = 45,148). Additionally, we excluded subjects with missing nutrient intake data in both the control group (*n* = 105) and in patients with NAFLD (*n* = 27). Finally, 2950 patients with NAFLD and 12,907 healthy controls were included in the analyses (Figure 1). The baseline characteristics of the patients with NAFLD and healthy controls are summarized in Table 1. The NAFLD group exhibited more elements of metabolic syndrome, such as higher blood pressure, higher serum levels of glucose and cholesterol, and a higher waist circumference. In contrast, there were more current drinkers and smokers in the control group (both *p* < 0.001) than in the NAFLD group. 

The frequency of the *PNPLA3* rs738409 *G* allele was 0.43 in the total included population, while it was 0.47 and 0.42 in the NAFLD and control groups, respectively. In our study, the *PNPLA3* risk allele group (rs738409, *GG + GC*) had a higher proportion of NAFLD cases than the non-risk allele group (rs738409, *CC*) (19.9% [2110/10,616] vs. 16.0% [840/5241]; Table 1). The *PNPLA3* rs738409 *G* allele showed a significant association with NAFLD in all additive, dominant, and recessive model analyses (odds ratio [OR] of the additive model = 1.22; 95% confidence interval, 1.15–1.30; *p* = 1.96 × 10^−10^) (Appendix A). After adjusting for age, sex, alcohol intake, and smoking, the *PNPLA3* rs738409 *G* allele remained an independent risk factor for NAFLD (adjusted OR = 1.22; *p* = 1.75 × 10^−10^) (Table 2). 

### 3.1. Association between Nutrients and NAFLD in the Total Population Subsection

In the macronutrient-consumption assessments, the healthy control group contained a significantly larger proportion of subjects with high protein (*p* < 0.001) and fat (*p* < 0.001) intakes (Appendix A) compared to the NAFLD group. The proportion of subjects with a high carbohydrate intake was larger in the NAFLD group than in the control group, but this difference was not statistically significant (*p* = 0.067) (Appendix A). 

In the mineral- and vitamin-consumption assessments, a significantly larger proportion of subjects in the healthy control group had higher intakes of sodium (*p* = 0.003), phosphorus (*p* < 0.001), zinc (*p* = 0.04), vitamin B1 (*p* < 0.001), vitamin B2 (*p* < 0.047), niacin (*p* < 0.001), and vitamin B6 (*p* < 0.001) than those in the NAFLD group (Appendix A). 

Subsequently, the proportions of NAFLD cases were compared between the high- and low-nutrient-intake groups (Table 3). High intakes of protein (OR = 0.765, *p* < 0.0001), fat (OR = 0.616, *p* < 0.0001), sodium (OR = 0.884, *p* < 0.0001), phosphorus (OR = 0.850, *p* < 0.0001), zinc (OR = 0.919, *p* = 0.043), vitamin B1 (OR = 0.771, *p* < 0.0001), vitamin B2 (OR = 0.874, *p* = 0.044), niacin (OR = 0.746, *p* < 0.0001), and vitamin B6 (OR =0.842, *p* < 0.0001) significantly decreased the risk of NAFLD.

### 3.2. Association between Nutrients and NAFLD According to the PNPLA3 Genotype Figures, Tables, and Schemes

To investigate the differential association between nutrient intake and NAFLD according to the *PNPLA3* genotype, the population was divided into two groups (Table 4). In the *PNPLA3* risk allele group (rs738409, *GG + GC*), high intakes of protein (OR 0.821, *p* = 0.001), fat (OR 0.755, *p* < 0.0001), sodium (OR 0.771, *p* < 0.0001), phosphorus (OR 0.851, *p* = 0.009), niacin (OR 0.800, *p* < 0.0001), vitamin B6 (OR 0.823, *p* = 0.001), and ash (OR 0.833, *p* = 0.017) were associated with a decreased risk of NAFLD (Table 4). In the non-risk allele group (rs738409, *CC*), a high fat intake was associated with a decreased risk of NAFLD (OR 0.794, *p* = 0.009), but other nutrients did not show a significant association (Table 4). 

Subsequently, the interaction between nutrients and the *PNPLA3* genotype was analyzed. Among the nutrients investigated, only a high sodium intake had a statistically significant protective interaction with the *PNPLA3* genotype against NAFLD development (interaction *p* = 0.002) (Table 4). The data are described in more detail in Appendix A and Figure 2. In these univariate analyses, a high intake of sodium exhibited a significant protective effect against NAFLD in the *PNPLA3* risk group (*p* < 0.0001) (Appendix A), while no such effect was observed in the *PNPLA3* non-risk group (*p* = 0.851) (Appendix A). 

### 3.3. Protective Foods against NAFLD in the PNPLA3 Risk Allele Group

To determine which food the sodium originated from, the association between food-intake frequency and NAFLD was investigated in the *PNPLA3* genotype groups (Table 5). In the *PNPLA3* risk allele group (rs738409, *GG + GC*), white rice, baechu kimchi, leaf mustard or scallion kimchi, green pepper, orange, coffee, sugar (for tea/coffee), and cream (for tea/coffee) were protective against NAFLD (all *p* < 0.05), while multigrain rice, yogurt, nuts, pickled radish, vegetable salad, and fried food increased the risk of NAFLD development (all *p* < 0.05). In the *PNPLA3* non-risk allele group (rs738409, *CC*), white rice, baechu kimchi, green pepper, orange, strawberry, pear, coffee, sugar (for tea/coffee), and cream (for tea/coffee) were protective (all *p* < 0.05), whereas multigrain rice, yogurt, green tea, and fried food increased the NAFLD risk (all *p* < 0.05).

Among the foods studied, baechu kimchi had a statistically significant protective interaction with the *PNPLA3* genotype against NAFLD development (*p* = 0.012) (Table 5 and Figure 2). This interaction between baechu kimchi and the *PNPLA3* genotype was very similar to the interaction between sodium and the *PNPLA3* genotype. Based on these observations, we concluded that the protective effect of a high sodium intake was probably due to the intake of baechu kimchi.

## 4. Discussion

Using the KoGES database, we discovered that nutrients and foods differentially affect the risk of NAFLD depending on the *PNPLA3* genotype, and an interaction exists between nutrients or foods and the *PNPLA3* genotype in the risk of NAFLD. A high fat intake decreased the risk of NAFLD, regardless of the *PNPLA3* genotype. High intakes of protein, phosphorus, sodium, niacin, vitamin B6, and carotene were associated with a decreased risk of NAFLD in the *PNPLA3* risk allele group. Among these protective nutrients, only sodium showed a significant interaction with the *PNPLA3* genotype. To identify the source of the sodium, the association between food intake frequency and the *PNPLA3* genotype was investigated. Only baechu kimchi, a salty food, had a significant protective interaction with the *PNPLA3* genotype with respect to high sodium intake: a high intake of baechu kimchi significantly decreased the risk of NAFLD, with a greater magnitude in the *PNPLA3* risk allele group than in the *PNPLA3* non-risk allele group. Therefore, the preventive effect of a high sodium intake in the *PNPLA3* risk allele group appears to be derived from high intake of baechu kimchi. 

In the current study, a high sodium intake was inversely associated with NAFLD. This observation was a conundrum. It is possible that a high-sodium diet may indeed have a preventive effect on NAFLD. A recent animal study demonstrated that mice fed a high-sodium and high-fat diet developed less hepatic steatosis, metabolic syndrome, and insulin resistance compared with those fed a normal or low-sodium and high-fat diet [31]. The reduced steatosis was associated with lower serum aldosterone levels and downregulation of hepatic mineralocorticoid receptors; thus, decreased activation of hepatic mineralocorticoids may have resulted in beneficial downstream inhibition of lipogenesis [31]. However, many studies have reported the opposite: a high-salt diet has been linked to increased glucocorticoid production, insulin resistance, metabolic syndrome, and NAFLD development [32,33,34]. To draw a reasonable explanation for our results, we searched for food items that correlated with a high sodium intake. Among the salty foods we investigated, salted seafood, cheese, and pickled radish increased the risk of NAFLD. Kimchi alone exerted a protective effect. Thus, we concluded that the major source of dietary sodium in this study was kimchi [35], and that the high sodium intake may have been a confounding factor. 

Kimchi, a representative Korean food, is a salted and fermented vegetable usually prepared from winter cabbage [36]. The health benefits of kimchi have been widely reported [36]. Among these is an improvement in metabolic markers such as blood glucose level, cholesterol level, insulin resistance, body weight, and body mass index, which have been tested in both animals [37,38,39] and humans [40,41,42]. Furthermore, kimchi exerts a beneficial effect against hepatic damage by reducing hepatic lipid synthesis and inflammatory cytokines [39]. These beneficial effects of kimchi on metabolic markers result from their modulation of gut microbiota [43,44,45,46]. Tests performed on probiotics extracted from kimchi were found to have the same beneficial effects [45,47,48,49].

The key factor linking kimchi and its protective effect against NAFLD may be the short-chain fatty acids (SCFAs) produced by the gut microbiome [50]. Among SCFAs, butyrate is known to reduce obesity and related metabolic complications, including NAFLD [50,51]. An animal study demonstrated that mice fed sodium butyrate and a high-fat diet had significantly decreased hepatic steatosis and decreased hepatic triglyceride and cholesterol levels compared with mice fed only a high-fat diet [52]. Another animal study demonstrated that SCFAs (acetate, propionate, and butyrate) ameliorate methionine- and choline-deficient diet-induced hepatic steatosis and inflammation [53]. A human study tested the gut microbiome and found fewer butyrate-producing bacteria in patients with NAFLD [54]. These studies indicated that SCFAs, including butyrate, exert a preventive effect against NAFLD. Probiotics (such as *Lactobacillus casei*) combined with plant extracts reduced the markers of NAFLD and increased the concentration of butyric acid in a mouse model [55]. As a fermented vegetable, kimchi simultaneously possesses the characteristics of both probiotics and prebiotics. Thus, it enhances gut microbial diversity [43] and, in turn, has a beneficial effect on NAFLD. In particular, the protective effect of kimchi on *PNPLA3* risk genotypes indicates that fermented vegetables, such as kimchi, may be chosen as tailored foods for those affected. 

In the present study, a high fat and protein intake had a preventive effect against NAFLD. A high-fat and low-carbohydrate diet, often known as a ketogenic diet, is a popular weight-reduction method. Many clinical trials have demonstrated that a ketogenic diet is effective in improving NAFLD [56,57,58,59,60,61,62]. A ketogenic diet decreases intrahepatic insulin resistance and, thus, serum insulin concentration [56,57,58,59,60,61,62]. This, in turn, increases the net hydrolysis of intrahepatic triglycerides, thus improving hepatic steatosis, inflammation, and fibrosis [56,57,58,59,60,61,62]. Similarly, a low-carbohydrate, high-protein diet also has metabolic benefits with regard to NAFLD. A human study demonstrated that an isocaloric low-carbohydrate diet with a high protein content decreased hepatic de novo lipogenesis, increased serum β-hydroxybutyrate concentrations (reflecting increased mitochondrial β-oxidation), and increased folate-producing *Streptococcus* and serum folate concentrations [63]. Furthermore, transcriptomic analysis revealed downregulation of the fatty acid synthesis pathway and upregulation of the folate-mediated one-carbon metabolism and fatty acid oxidation pathways [63]. Our study demonstrated that a high fat intake produces a protective effect regardless of the *PNPLA3* genotype, whereas a high intake of protein is more beneficial in the *PNPLA3* risk allele group. Nevertheless, caution should be exercised, as a ketogenic diet can elevate liver enzymes and worsen lipid profiles in patients with NAFLD [64,65]. Additionally, this diet seems less effective in premenopausal women [66]; its recommendation should therefore be carefully considered for these patients. 

*PNPLA3* encodes a calcium-independent triacylglycerol lipase that contains a phospholipase domain. Upon liver injury, *PNPLA3* expression is induced in hepatic stellate cells (HSCs) [67]. In the presence of *PNPLA3* I148M, HSCs become more proinflammatory and profibrogenic and produce less retinol, which potentially mediates the transition of HSCs into a myofibroblast-like phenotype [67,68]. Our findings suggest that a high retinol intake reduces the risk of NAFLD and may be relevant in this regard. Whether retinol intake prevents NAFLD in *PNPLA3* risk genotypes warrants further investigation. 

This study had several limitations. First, there may have been a recall bias in collecting data on food intake habits because of the self-reporting nature of the food-frequency questionnaire. Second, the identification of fatty liver depended solely on the medical records of the subjects. The nationwide database did not include the results of imaging studies such as sonography, computed tomography, or magnetic resonance imaging. Thus, a detailed evaluation of NAFLD severity was not possible. Third, the present study could not count the various beneficial nutrients such as antioxidants, polyunsaturated fatty acids, or amino acids. As these beneficial nutrients may directly attenuate the oxidative stress that results in hepatic fibrosis [14,69], future studies need to consider these detailed aspects of nutrition. 

In the present study, the associations and interactions between NAFLD, the *PNPLA3* genotype, and nutritional and dietary factors were investigated. The genetic risk of NAFLD conferred by *PNPLA3* in the Korean population was verified. In addition, we discovered some nutritional and dietary factors that can significantly influence the development of NAFLD, and these factors may interact with the *PNPLA3* genotype. Both increasing the proportion of fat and protein relative to that of carbohydrate and consuming fermented vegetables may help reduce NAFLD, regardless of the *PNPLA3* genotype. In *PNPLA3* risk genotypes, the role of a high-protein diet appears to be more important, as does meeting the daily requirement of vitamins and minerals. Furthermore, kimchi, a fermented vegetable, may induce an added protective effect in *PNPLA3* risk genotypes because of its gene–diet interaction. These results suggest that planning a therapeutic diet with high protein and fermented vegetables may be a valid strategy for those with *PNPLA3* risk genotypes and NAFLD, and warrants future interventional studies. Furthermore, in the near future, studies are needed on the mechanisms of interactions between various beneficial nutrients, *PNPLA3*, and NAFLD development. 

## 5. Conclusions

The *PNPLA3* genotype is differentially associated with nutritional and dietary factors. In particular, it interacts with kimchi, a fermented vegetable. This indicates that tailored nutritional therapy based on an individual’s genetic background may be a good strategy to prevent NAFLD, and fermented vegetables may serve as a therapeutic food for those with the *PNPLA3* risk allele. 

## Figures and Tables

**Figure 1 nutrients-15-00152-f001:**
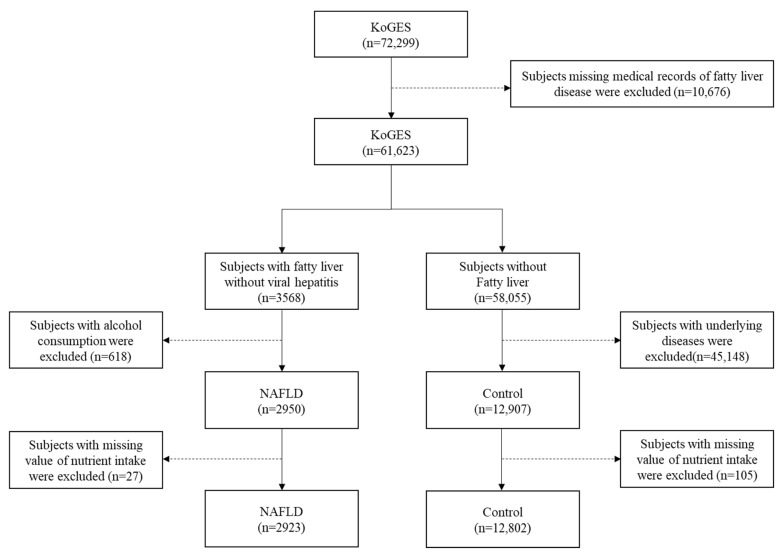
Study population using KoGES database. Abbreviations: KoGES, Korean Genome and Epidemiology Study; NAFLD, non-alcoholic fatty liver disease.

**Figure 2 nutrients-15-00152-f002:**
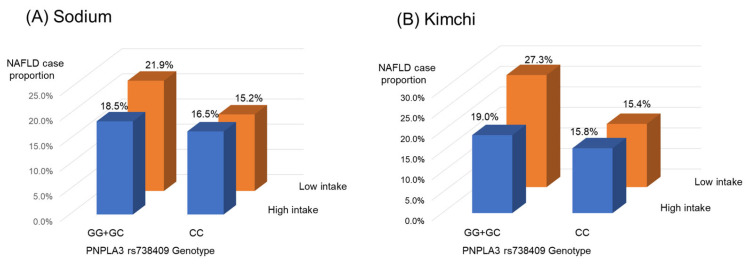
Impact of dietary sodium (**A**) and kimchi (**B**) intake on the genetic susceptibility of NAFLD. Abbreviations: NAFLD, non-alcoholic fatty liver disease; *PNPLA3*, patatin-like phospholipase domain-containing 3.

**Table 1 nutrients-15-00152-t001:** Baseline characteristics of the study population.

Characteristics	Total Population (*n* = 15,725)	NAFLD (*n* = 2923)	Control (*n* = 12,802)	*p*-Value
Age (years)	49.84 (±7.82)	55.77 (±7.58)	48.48 (±7.22)	0.01
Male (%)	7108 (45.20)	1481 (50.68)	5630 (43.98)	<0.001
Female (%)	8617 (54.80)	1442 (49.32)	7172 (56.02)	
Glucose (mg/dL)	92.09 (±15.03)	100.67 (±23.30)	90.12 (±11.51)	<0.001
Total cholesterol (mg/dL)	193.17 (±33.09)	197.11 (±37.00)	192.27 (±32.06)	<0.001
HDL cholesterol (mg/dL)	53.73 (±12.98)	48.49 (±11.37)	54.93 (±13.03)	<0.001
Triglyceride (mg/dL)	114.78 (±75.87)	153.08 (±100.12)	106.02 (±66.08)	<0.001
Waist circumference (cm)	80.26 (±8.55)	85.9 (±7.91)	78.97 (±8.16)	0.01
Systolic blood pressure (mmHg)	119.98 (±14.03)	125.68 (±14.46)	118.68 (±13.60)	<0.001
Diastolic blood pressure (mmHg)	74.84 (±9.58)	77.47 (±9.50)	74.24 (±9.50)	<0.001
Body mass index (kg/m^2^)	23.67 (±2.82)	25.53 (±2.88)	23.24 (±2.62)	0.01
Alcohol intake				<0.001
Non-drinker (*n* (%))	6823 (43.50)	1368 (46.92)	5455 (42.70)	
Ex-drinker (*n* (%))	579 (3.70)	206 (7.08)	373 (2.95)	
Current drinker (*n* (%))	8275 (52.80)	1340 (46.00)	6935 (54.35)	
Smoking				<0.001
Non-smoker (*n* (%))	10,590 (67.5)	1850 (63.39)	8740 (68.43)	
Ex-smoker (*n* (%))	2862 (18.3)	703 (24.13)	2159 (16.90)	
Current smoker (*n* (%))	2238 (14.3)	364 (12.47)	1874 (14.67)	
*PNPLA3* rs738409 genotype				
GG (*n* (%))	2911 (18.4)	652 (22.2)	2259 (17.5)	<0.001
GC (*n* (%))	7705 (48.6)	1458 (48.4)	6247 (48.4)	
CC (*n* (%))	5241 (33.0)	840 (28.4)	4401 (34.1)	

Values are mean (± standard deviation) or number (%). Abbreviations: *n*, number; NAFLD, non-alcoholic fatty liver disease; *PNPLA3*, patatin-like phospholipase domain-containing 3.

**Table 2 nutrients-15-00152-t002:** Genetic risk of *PNPLA3* rs738409 *G* allele on non-alcoholic fatty liver disease (NAFLD) occurrence in multivariable logistic regression analyses.

	NAFLD (*n*)	Control (*n*)	Proportion of NAFLD (%)	Additive Model	
				OR (95% CI)	*p-*Value *
*PNPLA3* rs738409 allele					
*C allele*	837	4380	16.04	1	
*G allele*	2100	8452	19.90	1.22 (1.15~1.30)	1.75 × 10^−10^
Sex					
Male	1485	5652	20.81	1	
Female	1452	7180	16.82	0.80 (0.75–0.85)	4.79 × 10^−11^
Age <49 years old >=49 years old	521 2416	7430 5402	6.55 30.9	1 4.68 (4.28–5.11)	5.8 × 10^−280^
Alcohol					
Non-drinker	1379	5479	20.11	1	
Ex-drinker	208	378	35.49	1.76 (1.56–1.98)	1.77 × 10^−16^
Current drinker	1350	6975	16.22	0.81 (0.75–0.86)	7.26 × 10^−10^
Smoking					
Non-smoker	1863	8781	17.5	1	
Ex-smoker	708	2169	24.61	1.40 (1.30–1.52)	2.9 × 10^−17^
Current smoker	366	1882	16.28	0.93 (0.84–1.03)	0.177

Abbreviations: NAFLD, non-alcoholic fatty liver disease; 95% CI, 95% confidence interval; OR, odds ratio; *PNPLA3*, patatin-like phospholipase domain-containing 3. * The *p*-values were calculated with R Statistical Software (v 4.1.2; R Core Team 2021).

**Table 3 nutrients-15-00152-t003:** Comparison of the non-alcoholic fatty liver disease (NAFLD) case proportions between several nutrients’ high-intake and low-intake groups among the * total participants (*n* = 15,725), as determined by univariate analyses.

Nutrient	High-Intake Group **			Low-Intake Group **			OR (95% CI)	*p-*Value
	NAFLD (*n*)	Control (*n*)	Proportion of NAFLD (%)	NAFLD (*n*)	Control (*n*)	Proportion of NAFLD (%)		
Energy	786	3823	17.05%	2137	8979	19.22%	0.864 (0.789–0.945)	0.001
Carbohydrate	2354	10,113	18.88%	569	2689	17.46%	1.10 (0.994–1.217)	0.064
Protein	1396	6971	16.68%	1527	5831	20.75%	0.765 (0.706–0.829)	<0.0001
Fat	1294	7209	15.22%	1629	5593	22.56%	0.616 (0.568–0.668)	<0.0001
Sodium	1678	7732	17.83%	1245	5070	19.71%	0.884 (0.815–0.959)	<0.0001
Potassium	285	1281	18.20%	2638	11,521	18.63%	0.972 (0.849–1.112)	0.677
Calcium	353	1508	18.97%	2570	11,294	18.54%	1.029 (0.909–1.164)	0.654
Phosphorus	2021	9282	17.88%	902	3520	20.40%	0.850 (0.778–0.927)	<0.0001
Iron	342	1460	18.98%	2581	11,342	18.54%	1.029 (0.908–1.167)	0.651
Zinc	1190	5475	17.85%	1733	7327	19.13%	0.919 (0.847–0.997)	0.043
Vitamin A	493	2160	18.58%	2430	10,642	18.59%	1.00 (0.898–1.113)	0.994
Carotene	1547	7026	18.05%	1376	5776	19.24%	0.924 (0.853–1.002)	0.055
Vitamin B1	383	2094	15.46%	2540	10,708	19.17%	0.771 (0.686–0.867)	<0.0001
Vitamin B2	302	1491	16.84%	2621	11,311	18.81%	0.874 (0.767–0.996)	0.044
Niacin	1052	5504	16.05%	1871	7298	20.41%	0.746 (0.686–0.81)	<0.0001
Vitamin B6	1553	7344	17.46%	1370	5458	20.06%	0.842 (0.777–0.913)	<0.0001
Folate	209	820	20.31%	2714	11,982	18.47%	1.125 (0.961–1.317)	0.142
Vitamin C	1265	5585	18.47%	1658	7217	18.68%	0.986 (0.909–1.069)	0.732
Vitamin E	395	1870	17.44%	2528	10,932	18.78%	0.913 (0.813–1.027)	0.129
Ash	491	2171	18.44%	2432	10,631	18.62%	0.989 (0.888–1.101)	0.835
Cholesterol	352	1715	17.03%	2571	11,087	18.82%	0.885 (0.783–1.0)	0.051

Abbreviations: NAFLD, non-alcoholic fatty liver disease; OR, odds ratio. * The proportion of NAFLD patients in the total population was 18.59% (2923/15,725). ** For each nutrient, daily intake above the recommended daily value was defined as high intake and below that as low intake. The recommended daily values used in this study are presented in Appendix A.

**Table 4 nutrients-15-00152-t004:** Association and interaction between nutrients and the *PNPLA3* genotype for the risk of non-alcoholic fatty liver disease (NAFLD) development. Multivariate logistic regression analyses adjusted for age, sex, and smoking were applied.

Food	*rs738409 GG + GC* (*n* = 10,616)				rs738409 CC (*n* = 5241)				Interaction *p*
	Proportion of NAFLD (%)		OR (95% CI)	*p*	Proportion of NAFLD (%)		OR (95% CI)	*p*	
	High Intake	Low Intake			High Intake	Low Intake			
Energy	18	20.6	0.946 (0.835–1.071)	0.381	15.1	16.4	0.943 (0.778–1.141)	0.551	0.931
Carbohydrate	20.3	18.3	1.037 (0.901–1.194)	0.617	16	15.8	0.94 (0.757–1.171)	0.577	0.545
Protein	17.7	22.3	0.821 (0.734–0.918)	0.001	14.8	17.5	0.863 (0.727–1.026)	0.094	0.381
Fat	16.1	24.2	0.755 (0.675–0.845)	<0.0001	13.4	19.1	0.794 (0.667–0.945)	0.009	0.154
Sodium	18.5	21.9	0.771 (0.689–0.863)	<0.0001	16.5	15.2	1.017 (0.853–1.214)	0.851	0.002
Potassium	19.8	19.9	0.991 (0.821–1.191)	0.922	15.2	16.1	0.901 (0.678–1.186)	0.466	0.508
Calcium	20.2	19.8	0.997 (0.837–1.183)	0.972	16.6	15.9	0.954 (0.736–1.226)	0.715	0.626
Phosphorus	18.9	22.3	0.851 (0.755–0.96)	0.009	15.9	16.4	0.971 (0.801–1.181)	0.77	0.11
Iron	19.4	19.9	0.98 (0.822–1.164)	0.817	18.1	15.7	1.226 (0.948–1.576)	0.115	0.284
Zinc	19	20.5	0.948 (0.847–1.061)	0.354	15.5	16.4	0.917 (0.771–1.091)	0.331	0.758
Vitamin A	19.3	20	0.905 (0.779–1.05)	0.19	17.2	15.7	1.075 (0.861–1.337)	0.517	0.208
Vitamin B1	16.1	20.6	0.881 (0.748–1.033)	0.122	14.1	16.4	0.889 (0.697–1.128)	0.339	0.801
Vitamin B2	18.2	20.1	0.959 (0.8–1.145)	0.649	14.3	16.2	0.819 (0.621–1.07)	0.15	0.696
Niacin	16.9	21.9	0.8 (0.713–0.897)	<0.0001	14.3	17.2	0.843 (0.706–1.005)	0.057	0.275
Vitamin B6	18.5	21.6	0.823 (0.736–0.92)	0.001	15.3	16.9	0.865 (0.728–1.028)	0.099	0.596
Folate	20.4	19.8	0.961 (0.765–1.2)	0.73	20.2	15.7	1.325 (0.968–1.796)	0.074	0.255
Vitamin C	19.8	20	0.97 (0.867–1.085)	0.594	15.8	16.1	1.016 (0.854–1.208)	0.859	0.862
Vitamin E	18.6	20.1	1.016 (0.864–1.191)	0.849	15.1	16.2	0.985 (0.77–1.251)	0.9	0.869
Ash	19.1	20	0.833 (0.716–0.967)	0.017	17.2	15.7	1.032 (0.826–1.283)	0.782	0.449

Abbreviations: *n*, number; NAFLD, non-alcoholic fatty liver disease; 95% CI, 95% confidence interval; OR, odds ratio; *PNPLA3*, patatin-like phospholipase domain-containing 3.

**Table 5 nutrients-15-00152-t005:** Association and interaction between foods and the *PNPLA3* genotype for the risk of non-alcoholic fatty liver disease (NAFLD) development. Multivariate logistic regression analyses adjusted for age, sex, and smoking were applied.

Food	*rs738409 GG + GC* (*n* = 10,616)				rs738409 CC (*n* = 5241)				Interaction *p*
	Proportion of NAFLD (%)		OR (95% CI)	*p*	Proportion of NAFLD (%)		OR (95% CI)	*p*	
	High Intake	Low Intake			High Intake	Low Intake			
Protective Foods									
White rice	15.07	21.13	0.785 (0.68–0.9)	0.001	10.69	17.3	0.674 (0.53–0.85)	0.001	0.689
Baechu kimchi	19.03	27.33	0.689 (0.57–0.83)	<0.001	15.79	15.4	0.745 (0.63–0.87)	<0.001	0.012
Leaf mustard or scallion kimchi	18.89	19.87	0.758 (0.62–0.94)	0.007	15.85	16	0.871 (0.71–0.99)	0.369	0.474
Green pepper	19.71	20.29	0.849 (0.73–0.99)	0.034	15.1	16.1	0.842 (0.74–0.96)	0.008	0.643
Orange	17.65	20.44	0.792 (0.66–0.95)	0.011	15.84	16.3	0.841 (0.72–0.98)	0.023	0.187
Strawberry	19.55	20.3	0.871 (0.75–1.02)	0.077	13.74	16.7	0.861 (0.76–0.98)	0.025	0.795
Pear	19.05	20.25	0.868 (0.41–0.67)	0.08	13.53	16.4	0.861 (0.47–0.71)	0.03	0.873
Coffee	17.62	26.25	0.743 (0.66–0.84)	<0.001	14.09	21.9	0.741 (0.67–0.82)	<0.001	0.487
Sugar for tea or coffee	17.64	22.68	0.751 (0.67–0.84)	<0.001	13.91	19.1	0.745 (0.68–0.82)	<0.001	0.757
Cream for tea or coffee	17.33	22.35	0.753 (0.68–0.84)	<0.001	13.8	18.1	0.761 (0.7–0.83)	<0.001	0.28
Risk Foods									
Multi grain rice	21.89	17.79	1.173 (1.06–1.3)	0.003	18.11	13.61	1.289 (1.09–1.52)	0.002	0.58
Yoghurt	23.55	19.49	1.159 (1–1.34)	0.045	19.04	15.3	1.147 (1.02–1.29)	0.025	0.59
Nuts (peanut, almond, pine nut)	26.54	19.34	1.284 (1.02–1.61)	0.032	23.02	15.44	1.297 (0.93–1.82)	0.131	0.703
Pickled radish	21.65	19.84	1.282 (1.03–1.59)	0.024	17.49	15.79	1.193 (0.85–1.67)	0.305	0.748
Vegetable salad	23.65	19.84	1.289 (1–1.66)	0.046	13.64	16.14	1.134 (0.91–1.41)	0.254	0.038
Green tea	19.56	19.89	1.135 (0.99–1.3)	0.065	17.15	15.59	1.149 (1.03–1.28)	0.014	0.554
Fried food	17.71	20.61	1.232 (1.10–1.38)	<0.001	14.34	16.54	1.254 (1.05–1.503)	0.014	0.745

Abbreviations: *n*, number; NAFLD, non-alcoholic fatty liver disease; 95% CI, 95% confidence interval; OR, odds ratio; *PNPLA3*, patatin-like phospholipase domain-containing 3.

## Data Availability

Not applicable.

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
