# Peer review of "Interaction between the PNPLA3 Gene and Nutritional Factors on NAFLD Development: The Korean Genome and Epidemiology Study"

_nutrients, 2022, doi:10.3390/nu15010152_

Round 1
Reviewer 1 Report
The paper presents data on gene-diet interactions in NAFLD, based on a meta-analyis of a known genetic risk factor (PNPLA3), in combination with dietary factors in a Korean cohort. The major outcome, that kimchi, a staple of southeast asian diet appears to favorably influence the incidence of NAFLD involving the PMPLA3 SNP risk factor is clear enough but limits the usefulness of the study, and may have covered less obvious influencing factors. In any case this particular finding highlights the need to identify causal factors due to the impact nutrition and different microbiome settings may have.
There are several points that should be addressed:
1) Dietary assessments based on self-reporting in food frequency questionnnaires are often problematic, in particular if the extend one year into the past. Were there checks applied regarding the correctness and credibility of the reports? Were checks applied to the self-reported smoking and drinking?
2) The statistical methods and the software used need to be reported in more detail. It would also be helpful for the reader to give some brief explanation on the methods used and their significance.
Note that lines 117 to 128, and 347-348, still contain the guidance text from the template - this should be removed.
3) In Table 2, the p-values given for the odds ratios reported appear extremely low. Since I am not familiar with this type of reporting, and probably most readers either, it might be good to have an explanation, and also, why are confidence intervals given only for one (or two) of the alternative possibilities?
4) similarly, confidence intervals are not given in Table 3, but are given in Table 4
5) Line 315: PNPLA3 in not a phospholipase, but a triacylglycerol lipase that contains a phospholipase domain.
Author Response
Dear Reviewer
We appreciate your time and insightful comments. We tried our best to come up with answers to your comments. In doing so, we thing the quality of the manuscript is much improved. Thank you so much for the opportunity to be reviewed by you.
1) Dietary assessments based on self-reporting in food frequency questionnnaires are often problematic, in particular if the extend one year into the past. Were there checks applied regarding the correctness and credibility of the reports? Were checks applied to the self-reported smoking and drinking?
Answer: Thank you. We agree the problem of food frequency questionnaires(FFQ) for it depends on the subjective memory. Even so, it is a widely used method to assess nutritional status. Using the data acquired with the FFQ, we have already published original articles several times. We have added the references. Thank you.
Line 110-111;
For the dietary assessment, a semi-quantitative food frequency questionnaire (FFQ)[21-23] involving 103 items was developed for the KoGES[23].
- Kwon, Y.J.; Kim, J.O.; Park, J.M.; Choi, J.E.; Park, D.H.; Song, Y.; Kim, S.J.; Lee, J.W.; Hong, K.W. Identification of Genetic Factors Underlying the Association between Sodium Intake Habits and Hypertension Risk. Nutrients 2020, 12, doi:10.3390/nu12092580.
- Kwon, Y.J.; Park, D.H.; Choi, J.E.; Lee, D.; Hong, K.W.; Lee, J.W. Identification of the interactions between specific genetic polymorphisms and nutrient intake associated with general and abdominal obesity in middle-aged adults. Clin Nutr 2022, 41, 543-551, doi:10.1016/j.clnu.2021.12.040.
- Lee, S.B.; Choi, J.E.; Park, B.; Cha, M.Y.; Hong, K.W.; Jung, D.H. Dyslipidaemia-Genotype Interactions with Nutrient Intake and Cerebro-Cardiovascular Disease. Biomedicines 2022, 10, doi:10.3390/biomedicines10071615.
2) The statistical methods and the software used need to be reported in more detail. It would also be helpful for the reader to give some brief explanation on the methods used and their significance.
Answer: Thank you. We supplemented the methods with regards to statistical methods and the software used.
Line 138-141;
All genetic association tests were conducted using PLINK version 1.9 (https://www.cog‐ge-nom-ics.org/plink)[26]. Further, the phenotype characteristics were analyzed by the SPSS (IBM SPSS Statistics Inc., New York, NY, USA)[27] and R Statistical Software [28].
- Purcell, S.; Neale, B.; Todd-Brown, K.; Thomas, L.; Ferreira, M.A.; Bender, D.; Maller, J.; Sklar, P.; de Bakker, P.I.; Daly, M.J., et al. PLINK: a tool set for whole-genome association and population-based linkage analyses. Am J Hum Genet 2007, 81, 559-575, doi:10.1086/519795.
- Gray, C.D.; Kinnear, P.R. IBM SPSS statistics 19 made simple; Psychology Press: 2012.
- R Core Team, R. R: A language and environment for statistical computing. R foundation for statistical computing Vienna, Austria: 2018.
Note that lines 117 to 128, and 347-348, still contain the guidance text from the template - this should be removed.
Answer: Thank you so much. The parts are removed.
3) In Table 2, the p-values given for the odds ratios reported appear extremely low. Since I am not familiar with this type of reporting, and probably most readers either, it might be good to have an explanation, and also, why are confidence intervals given only for one (or two) of the alternative possibilities?
Answer: Thank you. We used R statistical software for this specific analysis. One reason for the low exponential value is because the R program gives out that low values as those are. The other reason is because epidemiologically, those factor with low p-values are indeed very strongly associated with NAFLD. Anyhow, the strong associations are consistent with the results in Table 1, though the results were mostly abbreviated as “<0.001” in the Table 1.
We changed the Table 2 to show the contrast of the actual comparing numbers. And added a comment on the footnote regarding the statistical program.
Line 171-175;
Table 2. Genetic risk of PNPLA3 rs738409 G allele on non-alcoholic fatty liver disease (NAFLD) occurrence in multivariable logistic regression analyses.
NAFLD(n) |
Control(n) |
Proportion of NAFLD(%) |
Additive model |
||
OR (95% CI) |
P – value* |
||||
PNPLA3 rs738409 allele |
|
|
|
|
|
C allele |
837 |
4380 |
16.04% |
1 |
|
G allele |
2100 |
8452 |
19.90% |
1.22 (1.15~1.30) |
1.75×10-10 |
Sex |
|
|
|
|
|
Male |
1485 |
5652 |
20.81% |
1 |
|
Female |
1452 |
7180 |
16.82% |
0.80(0.75-0.85) |
4.79×10-11 |
Age < 49 years old >=49 years old |
521 2416 |
7430 5402 |
6.55% 30.9% |
1 4.68 (4.28 – 5.11) |
5.8×10-280 |
Alcohol |
|
|
|
|
|
Non-drinker |
1379 |
5479 |
20.11% |
1 |
|
Ex-drinker |
208 |
378 |
35.49% |
1.76(1.56-1.98) |
1.77×10-16 |
Current drinker |
1350 |
6975 |
16.22% |
0.81 (0.75-0.86) |
7.26×10-10 |
Smoking |
|
|
|
|
|
Non-smoker |
1863 |
8781 |
17.5% |
1 |
|
Ex-smoker |
708 |
2169 |
24.61% |
1.40 (1.30-1.52) |
2.9×10-17 |
Current smoker |
366 |
1882 |
16.28% |
0.93(0.84-1.03) |
0.177 |
Abbreviations: NAFLD, non-alcoholic fatty liver disease; 95% CI, 95% confidence interval; OR, odds ratio; PNPLA3, patatin-like phospholipase domain-containing 3. *The P-values were calculated with R Statistical Software.
4) similarly, confidence intervals are not given in Table 3, but are given in Table 4
Answer: Thank you. We have added the confidence invervals.
Line 193-195;
Table 3. Comparison of the non-alcoholic fatty liver disease (NAFLD) case proportions between several nutrient’s high intake and low intake groups among the *total participants (n = 15,725) as determined by univariate analyses.
Nutrient |
High intake group** |
Low intake group** |
OR(95% CI) |
P-value |
|||||
NAFLD (n) |
Control (n) |
Proportion of NAFLD (%) |
NAFLD(n) |
Control (n) |
Proportion of NAFLD (%) |
||||
Energy |
786 |
3823 |
17.05% |
2137 |
8979 |
19.22% |
0.864(0.789-0.945) |
0.001 |
|
Carbohydrate |
2354 |
10113 |
18.88% |
569 |
2689 |
17.46% |
1.10(0.994-1.217) |
0.064 |
|
Protein |
1396 |
6971 |
16.68% |
1527 |
5831 |
20.75% |
0.765(0.706-0.829) |
<0.0001 |
|
Fat |
1294 |
7209 |
15.22% |
1629 |
5593 |
22.56% |
0.616(0.568-0.668) |
<0.0001 |
|
Sodium |
1678 |
7732 |
17.83% |
1245 |
5070 |
19.71% |
0.884(0.815-0.959) |
<0.0001 |
|
Potassium |
285 |
1281 |
18.20% |
2638 |
11521 |
18.63% |
0.972(0.849-1.112) |
0.677 |
|
Calcium |
353 |
1508 |
18.97% |
2570 |
11294 |
18.54% |
1.029(0.909-1.164) |
0.654 |
|
Phosphorus |
2021 |
9282 |
17.88% |
902 |
3520 |
20.40% |
0.850(0.778-0.927) |
<0.0001 |
|
Iron |
342 |
1460 |
18.98% |
2581 |
11342 |
18.54% |
1.029(0.908-1.167) |
0.651 |
|
Zinc |
1190 |
5475 |
17.85% |
1733 |
7327 |
19.13% |
0.919(0.847-0.997) |
0.043 |
|
Vitamin A |
493 |
2160 |
18.58% |
2430 |
10642 |
18.59% |
1.00(0.898-1.113) |
0.994 |
|
Carotene |
1547 |
7026 |
18.05% |
1376 |
5776 |
19.24% |
0.924(0.853-1.002) |
0.055 |
|
Vitamin B1 |
383 |
2094 |
15.46% |
2540 |
10708 |
19.17% |
0.771(0.686-0.867) |
<0.0001 |
|
Vitamin B2 |
302 |
1491 |
16.84% |
2621 |
11311 |
18.81% |
0.874(0.767-0.996) |
0.044 |
|
Niacin |
1052 |
5504 |
16.05% |
1871 |
7298 |
20.41% |
0.746(0.686-0.81) |
<0.0001 |
|
Vitamin B6 |
1553 |
7344 |
17.46% |
1370 |
5458 |
20.06% |
0.842(0.777-0.913) |
<0.0001 |
|
Folate |
209 |
820 |
20.31% |
2714 |
11982 |
18.47% |
1.125(0.961-1.317) |
0.142 |
|
Vitamin C |
1265 |
5585 |
18.47% |
1658 |
7217 |
18.68% |
0.986(0.909-1.069) |
0.732 |
|
Vitamin E |
395 |
1870 |
17.44% |
2528 |
10932 |
18.78% |
0.913(0.813-1.027) |
0.129 |
|
Ash |
491 |
2171 |
18.44% |
2432 |
10631 |
18.62% |
0.989(0.888-1.101) |
0.835 |
|
Cholesterol |
352 |
1715 |
17.03% |
|
2571 |
11087 |
18.82% |
0.885(0.783-1.0) |
0.051 |
5) Line 315: PNPLA3 in not a phospholipase, but a triacylglycerol lipase that contains a phospholipase domain.
Answer: Thank you. It was corrected accordingly.
Line 325-326;
PNPLA3 encodes a calcium-independent triacylglycerol lipase that contains a phospholipase domain.
Reviewer 2 Report
The authors present very interesting results regarding the interaction between a gene (PNPLA3), nutritional factors and NAFLD. The methodology is sufficient and the results support the discussion. However, I do have some comments.
I. Major comments:
1. In the introduction I suggest including a brief paragraph regarding the dietary factors related to the development of NAFLD, in addition to referring to oxidative stress and inflammation, as relevant aspects in the development of this pathology. PMID: 34687092; PMID: 26512643.
2. Regarding the analysis of nutrients, I suggest including a specific analysis of fatty acids (SFA, MUFA, n-6 PUFAs and n-3 PUFAs "EPA and DHA").
3. I suggest writing a brief paragraph regarding the clinical projections of the study, especially interventions to be carried out.
II. Minor comments:
1. Improve the wording of the objective of the study.
2. Review the wording of some paragraphs.
Author Response
Dear Reviewer
We appreciate your time and insightful comments. We tried our best to come up with answers to your comments. In doing so, we thing the quality of the manuscript is much improved. Thank you so much for the opportunity to be reviewed by you.
Major comments:
In the introduction I suggest including a brief paragraph regarding the dietary factors related to the development of NAFLD, in addition to referring to oxidative stress and inflammation, as relevant aspects in the development of this pathology. PMID: 34687092; PMID: 26512643.
Answer: Thank you. We made a major revision of the introduction and elaborated more on the dietary importance in relation to genetic factors. Also, we discussed in the limitation about the oxidative stress and beneficial nutrients such as PUFA.
Line 63-73;
- Introduction
In these circumstances where genetic factors directly affect the pathogenesis, but no specific drug therapy exist, the role of diet becomes crucial. So far, only a few studies sug-gested gene-diet interactions in NAFLD development. One study discovered that high in-take of carbohydrates, isoflavones, and methionine increased the risk of hepatic fibrosis in NAFLD in a PNPLA3 genotype-dependent manner[12]. Another study showed that the minor allele of the haplotype in the 22q13 loci increased the risk of NAFLD, which inter-acted with high carbohydrate intake including high consumption of noodles and meat[13]. As such, it was indicated that certain dietary patterns are better to be avoided in individu-als with risk genotypes, and that certain dietary patterns could potentially attenuate the genetic influence. These findings justify the idea of precision nutrition, a diet tailored to the individual genetic make-up[10,12-15].
- Hernandez-Rodas, M.C.; Valenzuela, R.; Videla, L.A. Relevant Aspects of Nutritional and Dietary Interventions in Non-Alcoholic Fatty Liver Disease. Int J Mol Sci 2015, 16, 25168-25198, doi:10.3390/ijms161025168.
Line 332-341;
- Discussion
Third, the present study could not count in the various beneficial nutrients such as antioxidants, polyunsaturated fatty acids or aminoacids. For these beneficial nutrients may directly attenuate the oxidative stress that results in hepatic fibrosis[14,67], future studies need to take into account those detailed aspects of nutrition.
- Videla, L.A.; Valenzuela, R. Perspectives in liver redox imbalance: Toxicological and pharmacological aspects underlying iron overloading, nonalcoholic fatty liver disease, and thyroid hormone action. Biofactors 2022, 48, 400-415, doi:10.1002/biof.1797.
Regarding the analysis of nutrients, I suggest including a specific analysis of fatty acids (SFA, MUFA, n-6 PUFAs and n-3 PUFAs "EPA and DHA").
Answer: Thank you. The current study did not have the data on the fatty acids. So, it was commented in the limitation.
Line 332-341;
- Discussion
Third, the present study could not count in the various beneficial nutrients such as antioxidants, polyunsaturated fatty acids or aminoacids. For these beneficial nutrients may directly attenuate the oxidative stress that results in hepatic fibrosis[14,67], future studies need to take into account those detailed aspects of nutrition.
- Videla, L.A.; Valenzuela, R. Perspectives in liver redox imbalance: Toxicological and pharmacological aspects underlying iron overloading, nonalcoholic fatty liver disease, and thyroid hormone action. Biofactors 2022, 48, 400-415, doi:10.1002/biof.1797.
I suggest writing a brief paragraph regarding the clinical projections of the study, especially interventions to be carried out.
Answer: Thank you. The last part of the discussion was modified accordingly.
Line 352-357;
These results suggest that planning a therapeutic diet constituting of high-protein and fermented vegetable for NAFLD with PNPLA3 risk genotypes may be a valid strategy and warrants future interventional studies. Furthermore, studies are needed on the mechanism of interaction between various beneficial nutrients and the PNPLA3 gene effects on NAFLD development in the near future.
Minor comments:
Improve the wording of the objective of the study.
Answer: Thank you. The introduction was revised extensively. The last paragraph was rewritten.
Line 83-88;
In this study, we hypothesized that a distinctive gene-diet interaction may exist between PNPLA3 and NAFLD. We planned to test this interaction in the Korean population: the majority of Koreans belongs to one ethnicity, called han minjok[19]. This endows genetic and dietary homogeneity that is beneficial in genetic and dietary studies. We used a large Korean cohort (Korean Genome and Epidemiology Study [KoGES] cohort) dataset obtained from the Korea Biobank Project (KBP) database.
Review the wording of some paragraphs.
Answer: Thank you. We have changed wordings during this revision. Further, we plan to undergo English editing once the revised text meets the reviewers’ expectation.
Reviewer 3 Report
Thank you that you give me opportunity to review this manuscript “ Interaction between the PNPLA3 gene and Nutritional factors on NAFLD development: the Korean Genome and Epidemiology Study”. The authors described a very interesting topic of the association and interaction between NAFLD, PNPLA3 genotype, and nutritional and dietary factors were investigated. The authors did a great job. I recommend the article should need minor corrections.
Some my comments on the manuscript are described below:
1. In Introduction
In the introduction, there is no information about the epidemiology of NAFLD in Korean society or about the Korean diet. Only in the discussion line 269 to 274 onwards is Kimchi, a representative Korean food..., Furthermore, kimchi exerts a beneficial effect against hepatic damage by reducing hepatic lipid synthesis and inflammatory cytokines.[22]
2. Material I Methods
2.4
Line 117-131
“The Materials and Methods should be described with sufficient details to allow others to replicate and build on the published results. Please note that the publication of your manuscript implicates that you must make all materials, data, computer code, and protocols associated with the publication available to readers. Please disclose at the submission stage any restrictions on the availability of materials or information. New methods and protocols should be described in detail while well-established methods can be briefly described and appropriately cited.
Research manuscripts reporting large datasets that are deposited in a publicly available database should specify where the data have been deposited and provide the relevant accession numbers. If the accession numbers have not yet been obtained at the time of submission, please state that they will be provided during review. They must be provided prior to publication.
Interventionary studies involving animals or humans, and other studies that require ethical approval, must list the authority that provided approval and the corresponding ethical approval code”
It looks like an instruction copied from somewhere. I don't understand this. What does this have to do with statistical analysis? Please explain.
3. Results
Line 137-140 the data given here do not agree with the data in table 1. e.g. 2,950 patients with NAFLD but in table 1 is 2,923, health controls 12,907 but in table 1 is 12,802 etc.... Please explain. Please check other data.
In Table 1 is missing data for women, it is only male.
|
4. Discussion
The discussion describes the Korean diet, which should rather be included in the introduction.
5. Conclusion
Line 347 .. This section is not mandatory but can be added to the manuscript if the discussion is unusually long or complex.
Again it looks like an instruction copied from somewhere. I don't understand this. What does this have to do with conclusions? Please explain.
Author Response
Dear Reviewer
We appreciate your time and insightful comments. We tried our best to come up with answers to your comments. In doing so, we thing the quality of the manuscript is much improved. Thank you so much for the opportunity to be reviewed by you.
- In Introduction
In the introduction, there is no information about the epidemiology of NAFLD in Korean society or about the Korean diet. Only in the discussion line 269 to 274 onwards is Kimchi, a representative Korean food..., Furthermore, kimchi exerts a beneficial effect against hepatic damage by reducing hepatic lipid synthesis and inflammatory cytokines.[22]
Answer: Thank you. The introduction was revised extensively. The commented was included in the introduction.
Line 74-82;
In Korea, the prevalence of NAFLD(32.87%-42.9%) out-numbered the global average(25.24%)[9,16]. Further, it is expected to increase in the future owing to the westernized diet, lack of exercise, and increase in obesity and type 2 diabetes[9]. Korean diet is based on steamed rice served with small side dishes made mainly with vegetables and less frequently with meat, poultry or fish, and it also uses fermented vegetable, Kimchi very frequently. Furthermore, traditional Korean diet used whole mixed grains and beans, thus, it was known to be healthy having a low glycemic index, low cholesterol, and high content of fibers[17]. However, traditional benefits have been overshadowed by excessive consumption of refined rice and westernization[18].
- Material I Methods
2.4
Line 117-131
“The Materials and Methods should be described with sufficient details to allow others to replicate and build on the published results. Please note that the publication of your manuscript implicates that you must make all materials, data, computer code, and protocols associated with the publication available to readers. Please disclose at the submission stage any restrictions on the availability of materials or information. New methods and protocols should be described in detail while well-established methods can be briefly described and appropriately cited.
Research manuscripts reporting large datasets that are deposited in a publicly available database should specify where the data have been deposited and provide the relevant accession numbers. If the accession numbers have not yet been obtained at the time of submission, please state that they will be provided during review. They must be provided prior to publication.
Interventionary studies involving animals or humans, and other studies that require ethical approval, must list the authority that provided approval and the corresponding ethical approval code”
It looks like an instruction copied from somewhere. I don't understand this. What does this have to do with statistical analysis? Please explain.
Answer: Thank you. Those were instructions embedded in the format, which was mistakenly unremoved during the manuscript preparation. We removed the part this time. Sorry about that.
- Results
Line 137-140 the data given here do not agree with the data in table 1. e.g. 2,950 patients with NAFLD but in table 1 is 2,923, health controls 12,907 but in table 1 is 12,802 etc.... Please explain. Please check other data.
Answer: Thank you. Participants with missing values in the nutrient intake was automatically excluded during the statistical analysis. We clarified the exclusion process in the text and the Figure 1.
Line 98-99;
- Materials and Methods
Subjects missing medical records of fatty liver disease were excluded.
Line 149-151;
- Results
Additionally, we excluded subjects with missing data of nutrient intake in controls (n=105) and patients with NAFLD (n=27).
Figure 1. Study population using KoGES database.
In Table 1 is missing data for women, it is only male.
Answer: Thank you. The data was added.
Table 1. Baseline characteristics of the study population.
Characteristics |
Total population (n=15,725) |
NAFLD (n=2,923) |
Control (n=12,802) |
p-value |
Age (years) |
49.84 (±7.82) |
55.77(±7.58) |
48.48(±7.22) |
0.01 |
Male (%) |
7,108 (45.20) |
1,481 (50.68) |
5,630 (43.98) |
<0.001 |
Female (%) |
8,617 (54.80) |
1,442 (49.32) |
7,172(56.02) |
|
- Discussion
The discussion describes the Korean diet, which should rather be included in the introduction.
Answer: Thank you. Korean diet was briefly introduced in the introduction, as was answered for your comment 1.
- Conclusion
Line 347 .. This section is not mandatory but can be added to the manuscript if the discussion is unusually long or complex.
Again it looks like an instruction copied from somewhere. I don't understand this. What does this have to do with conclusions? Please explain.
Answer: Thank you. Those were instructions embedded in the format, which was mistakenly unremoved during the manuscript preparation. We removed the part this time. Sorry about that.